# A Global Review of Progress in Remote Sensing and Monitoring of Marine Pollution

Jingwu Ma [1,2,3], Renfeng Ma [3,4,5,*], Qi Pan [4], Xianjun Liang [6], Jianqing Wang [7] and Xinxin Ni [8]

1   Key Laboratory of Ocean Space Resource Management Technology of Ministry of Natural Resources, Zhejiang Institute of Marine Sciences, Hangzhou 310012, China; majingwu12@163.com
2   Land Consolidation and Rehabilitation Center, Wenzhou Bureau of Natural Resources and Planning, Wenzhou 325027, China
3   Ningbo University Donghai Academy & Zhejiang Ocean Development Think Tank Alliance, Ningbo 315211, China
4   Department of Geography and Spatial Information Techniques, Ningbo University, Ningbo 315211, China; 2011087072@nbu.edu.cn
5   Zhejiang Collaborative Innovation Center & Ningbo Universities Collaborative Innovation Center for Land and Marine Spatial Utilization and Governance Research, Ningbo University, Ningbo 315211, China
6   Ningbo Junzhi Engineering and Environmental Consulting Co., Ltd., Ningbo 315211, China; nbuandrew@163.com
7   Ningbo Institute of Oceanography, Ningbo 315832, China; wangjq@nbio.org.cn
8   Department of International Hotel Management at Qiandaohu, Tourism College of Zhejiang China, Hangzhou 311231, China; nixinxin@tourzj.edu.cn
*   Correspondence: marenfeng@nbu.edu.cn

**Abstract:** With the rapid development of urbanization and industrialization, human activities have caused marine pollution in three ways: land source, air source, and sea source, leading to the problem of marine environments. Remote sensing, with its wide coverage and fast and accurate monitoring capability, continues to be an important tool for marine environment monitoring and evaluation research. This paper focuses on the three types of marine pollution, namely marine seawater pollution, marine debris and microplastic pollution, and marine air pollution. We review the application of remote sensing technology methods for monitoring marine pollution and identify the limitations of existing methods. Marine seawater pollution can be effectively monitored by remote sensing technology, especially where traditional monitoring methods are inadequate. For marine debris and microplastic pollution, the monitoring methods are still in the early stages of development and require further research. For marine air pollution, more air pollution parameters are required for accurate monitoring. Future research should focus on developing marine remote sensing with data, technology, and standard sharing for three-dimensional monitoring, combining optical and physical sensors with biosensors, and using multi-source and multi-temporal monitoring data. A marine multi-source monitoring database is necessary to provide an immediately available basis for coastal and marine governance, improve marine spatial planning, and help coastal and marine protection.

**Keywords:** marine environmental monitoring; governance; marine pollution; coastal zone management





## 1. Introduction

The ocean is an important environment for humans, and its various types and scales of currents result in the distribution of liquid and three-dimensional resources, which form an important and unique system of marine resources and environment. The ocean is abundant in biological resources, mineral resources, energy, and other resources, and it is attracting increasing attention from the academic circle and industry. The rapid development of industrialization and urbanization has led to a concentration of human activities along coastlines, resulting in the degradation of coastal bays, marine resources, and the environment [1].

Generally, the coastal natural shoreline and coastal mudflat wetlands have been continuously reduced, the area of mangroves and coral reefs has been greatly reduced, and the marine ecological environment has been polluted. The situation is becoming increasingly serious, with increasing eutrophication of seawater and frequent occurrences of marine ecological disasters such as brown tides, green tides, and red tides posing a serious threat to migratory waterfowl and marine biodiversity. Meanwhile, the above environmental pollution, biological extinction, and natural disasters have posed a threat to the sustainable development of the coastal and marine environment. Dissolved organic matter from sewage treatment plants, humus from farmland, anthropogenic shoreline erosion, and the removal of native vegetation can cause significant increases in turbidity in coastal waters [2]. For example, NASA satellite imagery has shown that the water quality of Florida's Tampa Bay decreases in the winter months compared to the summer. More particles suspended in the water, a measure called turbidity, show up as yellow, orange, and red in December (a) than in July (b) due to seasonal freshwater discharge from nearby rivers and runoff into the bay, which carry nutrients (Figure 1). Hence, marine pollution monitoring is of great significance in terms of both theoretical and practical value [3–5]. Marine environment monitoring is an essential step in maintaining the quality of the marine environment and securing its ecology, and it is crucial for achieving sustainable marine development.

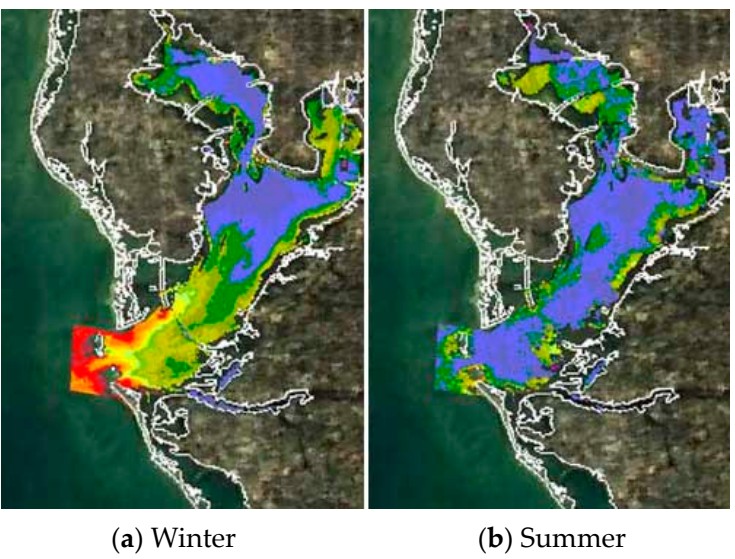

(**a**) Winter                    (**b**) Summer

**Figure 1.** Water quality of Florida's Tampa Bay decreases (Picture source NASA/USF).

Remote sensing plays a crucial role in marine monitoring, including the ecological monitoring of wetland, mangrove, coral reefs, and other key organisms (Figures 2 and 3), the environmental disaster monitoring of oil spills and red tides, the mapping of mesoscale coastlines, the measurement of chlorophyll-a, suspended sediment concentration, seawater temperature in offshore areas, and the automatic recognition of key targets such as dams, breeding areas, buildings, ports, bridges, and ships [6–8].

Based on a search of the Web of Science core collection from 2000 by using the keyword "marine remote sensing," it can be seen that marine remote sensing monitoring is a crucial aspect of marine ecological monitoring (Table 1). Remote sensing monitoring has been of interest to scholars from institutions that have published a number of important papers in this field, such as the Woods Hole Oceanographic Institution, the University of Washington, the University of California, San Diego, Oregon State University, the Plymouth Marine Laboratory, the First, Second, and Third Institutes of Oceanography, the Marine Environmental Monitoring Center, the Marine Technology Center, the Tianjin Marine Environmental Monitoring Center Station, the Ocean University of China, and Xiamen University in China. Meanwhile, government agencies around the world, such as NASA, NOAA, CNRS, and the Canadian Department of Fisheries and Oceans, invest heavily in research in this field.

Meanwhile, NASA, NOAA, and other institutes have established specific research directions in marine remote sensing technology for marine research and also collaborate with universities, whose research not only focuses on marine pollution monitoring, oil spills, dissolved organic carbon, coral issues, and thermal pollution [9–13], but also large-scale marine currents and air-sea interactions. For example, Chinese relevant institutes have specially established a research department in the field of marine remote sensing. The Second Institute of Oceanography of the Ministry of Natural Resources has taken the lead in conducting research on seawater color remote sensing in China and has achieved fruitful results in the mechanism and algorithm of seawater color remote sensing monitoring, as well as the development of domestically produced software system platforms. Relevant units such as the National Satellite Ocean Response Center and the North China Sea Administration of the Ministry of Natural Resources have all implemented operational monitoring applications for offshore oil spills. Furthermore, the Copernicus program also made a lot of contributions, whose program is the Earth observation component of the European Union's Space program, which offers information services that draw from satellite Earth observation and in-situ (non-space) data. Amounts of global data from satellites and ground-based, airborne, and seaborne measurement systems provide information to help service providers, public authorities, and other international organizations improve European citizens' quality of life and beyond. And organizations such as USEPA and US IOOS are monitoring exhaust from ships. University of Delaware professors have used remote sensing technology to monitor exhaust from ships.

**Table 1.** The top 10 related institutes of marine remote sensing monitoring and their research directions.

| Research Institute | Number of Paper | Country/ Region | Related Laboratory | Research Direction |
|---|---|---|---|---|
| National Aeronautics and Space Administration | 1506 | United States | / | Sea level rise monitoring; Marine ecosystem research; Marine currents motion research |
| National Oceanic and Atmospheric Administration | 1276 | United States | Pacific Marine Environment Laboratory | Climate-weather research; Marine ecosystem research; Ocean and coastal evolution research |
| Chinese Academy of Sciences | 1028 | China | Global Monitoring Laboratory | Greenhouse gases and carbon cycle research; Changes in clouds, aerosols and surface radiation research; Recovery of stratospheric ozone research |
| | | | Digital Earth Key Laboratory | Frontier theory and technology of earth observation research; Earth big data science and methods research; Digital earth science and platform research; Global environmental resources spatial information system research |
| | | | Key Laboratory of Infrared Detection and Imaging Technology | High-resolution infrared imaging technology research; Hyperspectral imaging technology research; Weak target detection technology research; High quantitative remote sensing detection technology research |
| California Institute of Technology | 853 | United States | Linde Center | Earth climate change research [14]; Pollution impact research; Carbon dioxide changes research |
| University of Washington | 617 | United States | Applied Physics Laboratory- Air-sea interaction and remote sensing | Sea-air exchange research; Coastal research; Sensor research; Wave research |

**Table 1.** *Cont.*

| Research Institute | Number of Paper | Country/ Region | Related Laboratory | Research Direction |
|---|---|---|---|---|
| University of Maryland | 576 | United States | Earth System Science Interdisciplinary Center (cooperation with NASA) | Climate variability and change research; Atmospheric composition and processes research; Global carbon cycle research; Global water cycle research |
| University of California, San Diego | 546 | United States | Scripps Institution of Oceanography | Collect and process data on the Earth, oceans and atmosphere by cameras, lasers and various electromagnetic sensors |
| University of Colorado | 476 | United States | Mortenson Center in Global Engineering | Sustainable water treatment system research; Field and remote sensing research; Infrastructure resilience and disaster recovery research |
| | | | Earth Science and Observation Center, Institute for Research in Environmental Sciences | Analysis of remote sensing data, validation of data |
| University of Miami | 445 | United States | Upper Ocean Dynamics Laboratory | Experimental studies on coastal circulation processes and ocean-atmosphere interactions during the hurricane |
| Woods Hole Oceanographic Institution | 385 | United States | Claisen Laboratory | Studies on air-sea interactions and their impact on weather and climate through a wide range of measured and remote sensing data |

Note: The laboratory information is compiled from the official websites of various organizations.

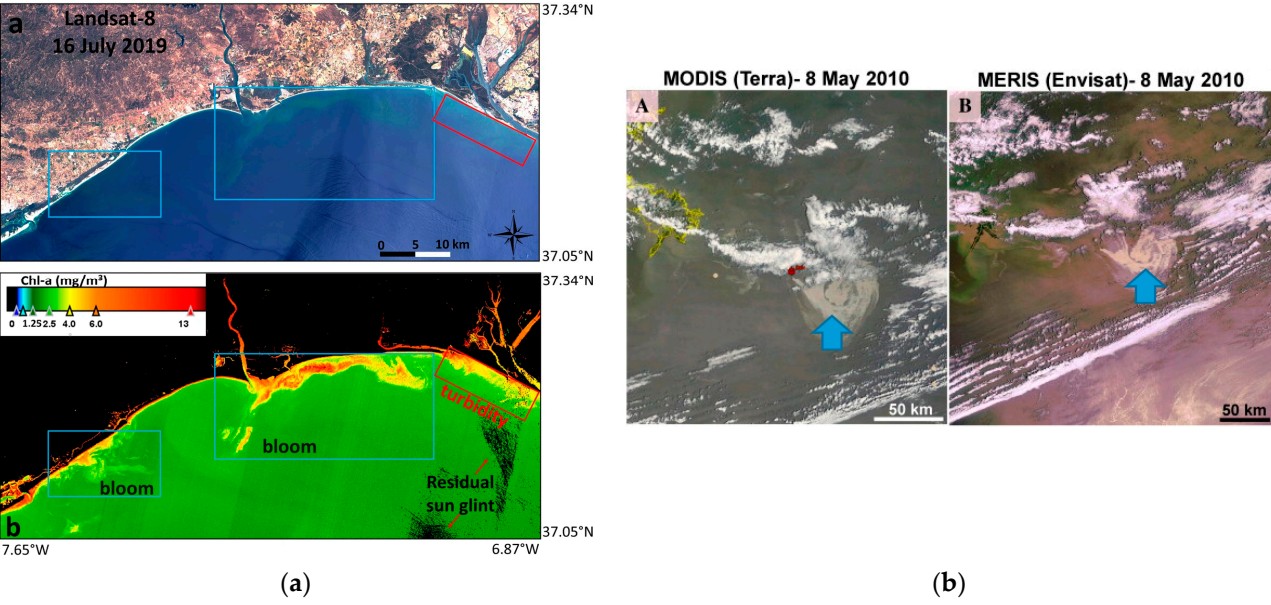

**Figure 2.** The polyedra algal bloom and oil spill of remote sensing images. (**a**) Landsat-8 RGB (bands 4-3-2) composite in this study region on 16 July 2019; (**b**) chl-a concentration (mg/m$^3$) after atmospheric correction with ACOLITE for the same scene [15]. (**A**) MODIS; (**B**) MERIS showing the site of the DWH oil spill in blue arrow [16].

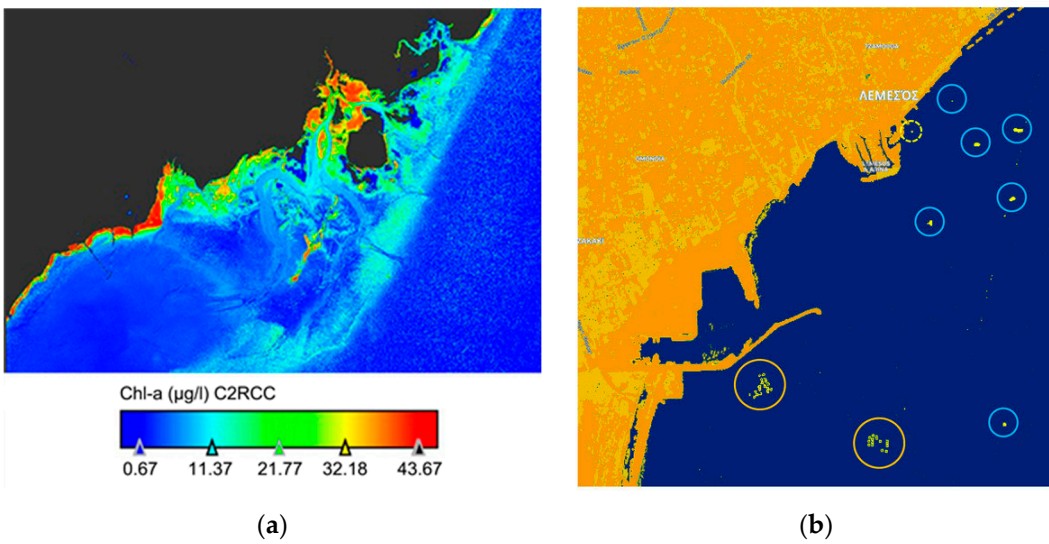

(**a**)                                                    (**b**)

**Figure 3.** The LiDAR and MSI remote sensing images (**a**) show the spatial distribution of Chl-a (conc_chl) C2RCC in the Kneiss Archipelago, Gulf of Gabes, Tunisia [17]. (**b**) the floating plastic litter from space using Sentinel-2 imagery [18].

Previous studies have reviewed the literature in the remote sensing field from the perspective of satellite functions and monitoring targets; however, they have not adequately reviewed the field of marine pollution monitoring. Therefore, a global review of progress in remote sensing monitoring of marine pollution is needed, which reviews the application domains of marine remote sensing technology and the progress in marine pollution monitoring (Figure 4), serving as a reference for academic research and development efforts in using remote sensing technology for marine pollution monitoring.

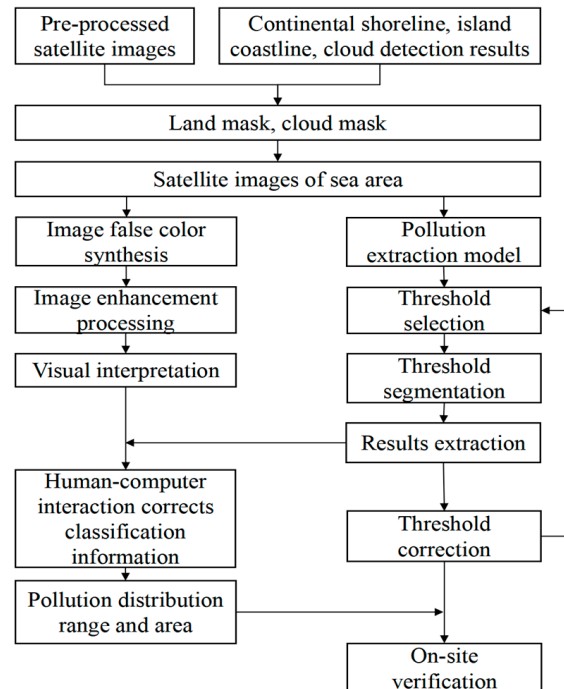

**Figure 4.** Framework of this study.

## 2. Sources and Monitoring Indices of Marine Pollution

The overexploitation of marine resources results in the degradation of the marine ecosystem and causes pollution, which damages the marine environment. The overexploitation, including major river conservation projects, land reclamation projects, coastal mining, offshore drilling, and mariculture, also leads to a shortage of marine resources and has negative impacts on coastal marine resources and environments. The utilization of marine resources varies in different regions due to differences in distribution, resulting in different types of pollution, such as land source, sea source and air source (Figure 5).

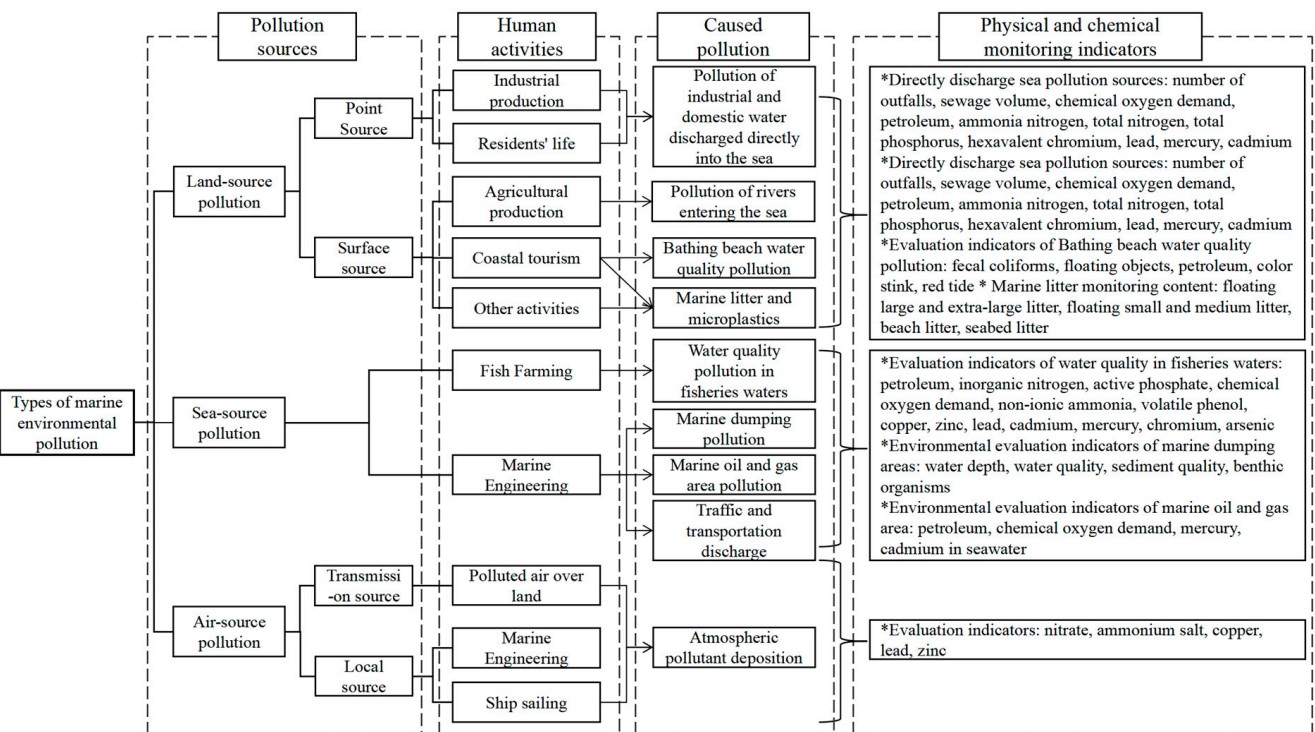

**Figure 5.** Source and monitoring index of marine pollution. Note: monitoring indices and methods from the Bulletin of China's Marine Ecological Environment in 2020.

**Land-source pollution** has the widest distribution and largest quantity, accounting for 80% of the total marine pollution and having the most serious effect. Land-source pollution can be divided into two types: point-source and area-source. Point-source pollution is primarily composed of industrial waste water and domestic sewage that is collected and treated through sewage networks and discharged into the sea through designated discharge points [19,20]. Area-source pollution comes from agricultural runoff and livestock breeding activities, which flow into the ocean through runoff [20]. The root cause of land source pollution is human economic and social activity, which contributes to the negative impacts of such activities on the marine environment [21]. This type of pollution is not limited to coastal areas but is influenced by economic and social activities and is more pronounced in offshore regions with intensive secondary industries and rapid economic development [22]. Additionally, human activities on land can cause eutrophication and biotoxicity in sea areas through river dams and water storage projects and reduce river sediment discharge, leading to coastal erosion and changes in river water and groundwater quality [23,24]. Coastal tourism, being a popular form of tourism, also contributes to marine pollution through the construction of tourist resorts, docks, and breakwaters, which fragment coastal habitats and harm biodiversity, and through the pollution of seawater quality and the flow of beach litter into the ocean from bathing beaches [25]. The water quality of direct discharge into the sea is the main evaluation index for monitoring point-source land source pollution, while the water quality of seaport rivers, beach water, marine debris, and microplastics is the

main evaluation index for monitoring area-source land source pollution. Water quality is primarily evaluated according to the "Standard for Seawater Quality" (GB 3097-1997) [26], the "Standard for Surface Water Environmental Quality" (GB 3838-2002) [27] for rivers flowing into the ocean, and the "Guide for Monitoring and Evaluation of Bathing Beaches" (HY/T 0276-2019) for bathing beaches.

**Sea-source pollution** mainly refers to the pollution caused by human utilization of the sea. Marine pollution offshore is primarily caused by fishing and aquaculture practices. Overfishing has resulted in a reduction of fish resources, leading to the implementation of non-standard practices such as beach aquaculture, offshore cage aquaculture, and pelagic fishing, which negatively impact the marine environment by releasing nutrients and drugs into the water [28]. This can also pose a threat to coastal biodiversity, affecting beaches and mangroves [29]. Another significant source of marine pollution is marine engineering activities, such as oil and gas field exploitation, which can result in oil spills, sewage discharge, and seepage from eroded oil-bearing rocks. Additionally, the unregulated discharge of waste water from cruise ships and cargo ships in coastal tourism contributes to marine pollution [30]. Monitoring marine pollution includes monitoring the water quality of fishery waters and the discharge of waste water from marine engineering activities. The quality of fishery waters is evaluated according to the "Fishery Water Quality Standard" (GB 11607-1989) [31], while monitoring of marine engineering pollution in dumping areas and oil and gas fields should be based on the "Technical Regulations for Monitoring of Marine Dumping Areas" and "Technical Guidelines for Environmental Impact Assessment of Marine Engineering" (GB/T 17108-2006) [32].

**Air-source pollution** can be divided into transmission sources and local sources (Figure 6). Transmission sources refer to the pollution in the air that is transported from the land to the sea due to the influence of monsoons. Local sources refer to the pollutants that are emitted directly into the sea, such as oil and gas field exploitation, marine transportation, and fishing. The pollutants include $SO_2$, NOx, CO, particulate matter, and VOCs. Atmospheric pollutants fall onto the marine surface through dry deposition and wet deposition processes [33]. Dry deposition involves the physical, chemical, and biological processes by which atmospheric pollutants fall onto the marine surface, while wet deposition involves ionic pollutants and soluble pollutants from the air falling onto the marine surface through precipitation or water vapor condensation [34]. The evaluation of marine atmospheric pollution deposition is mainly conducted based on the "Technical Regulations for the Assessment of the Flux of Atmospheric Pollutants Deposition into the Sea (Trial)" and includes the observation of the elements nitrate, $NH_4Cl$, Cu, Pb, and Zn.

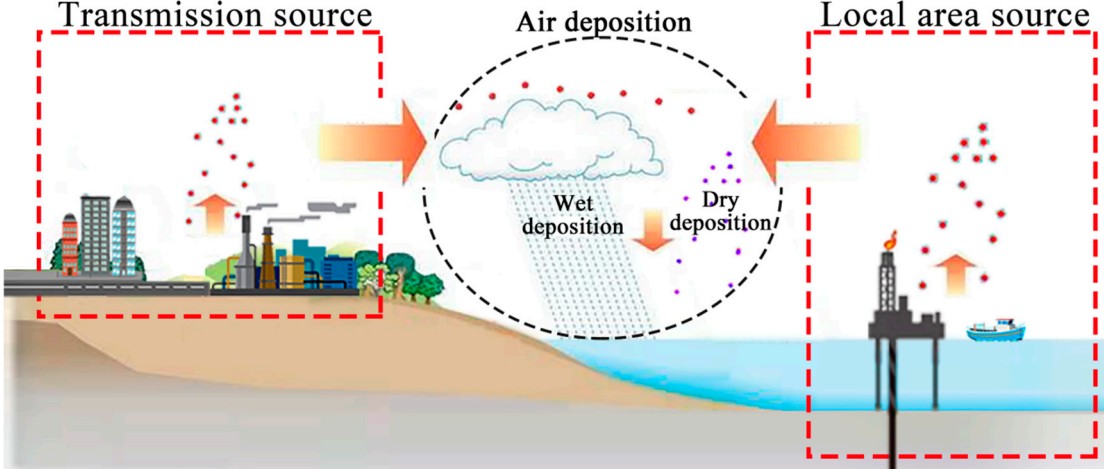

**Figure 6.** Mechanisms of marine atmospheric pollution transfer.

## 3. The Utilization of Remote Sensing Monitoring in Marine Pollution

### 3.1. An Overview of the Development History and Present Situation

In 1960, the United States launched the world's first meteorological satellite, TIROS-1, and used it to collect data on the marine surface temperature field from an altitude of approximately 700 km [35]. In 1978, the United States took the lead in the field of marine observation with the launch of Seasat-1, the first satellite dedicated to observing coastal seawater color [36], which marked the beginning of marine remote sensing monitoring. As the importance of marine resources continues to be recognized, more and more countries are investing in marine remote sensing technology to support monitoring efforts in the fields of shipping, environment, military, and economics [36–38].

The sensors commonly used in marine research include seawater color sensors, infrared sensors, microwave altimeters, microwave scatterometers, synthetic aperture radars, and microwave radiometers (Table 2). These sensors are capable of monitoring elements from the marine surface to the upper, lower, and bottom topography and can be used to monitor the seawater color and environment, as well as the marine dynamic environment (such as the marine surface wind field, marine surface height, effective wave height, and surface temperature) [39–41]. The sensors play a crucial role in monitoring islands, coastal zones, and other targets, including the marine wave field, storm surge floodplain, internal waves, sea ice, and oil spills. With the progress in microwave, infrared, and hyperspectral remote sensing technologies and their wider applications in the water, ecological, and air environments, the types of sensors available have become more diverse and specialized, and the spectral bands more refined. For example, China has launched a series of marine satellites, including HY-1C/D and HY-2B/C/D, equipped with new ultraviolet imagers, onboard calibration spectrometers, automatic ship identification systems (Figure 7), and other loads, providing a wealth of data for monitoring the global seawater color and dynamics [42]. The spectral bands of remote sensing images are getting finer and finer, forming multispectral, hyperspectral, and ultraspectral. At present, the spectral resolution in the 100 nm order is called multispectral, such as a remote sensor in the visible light and near-infrared spectral region only a few bands, such as the United States LandsatMSS, TM, SPOT in France, etc. The spectral resolution is in the 10 nm order, which is called HyPerspectral remote sensing. With the further improvement of the spectral resolution of remote sensing, remote sensing enters the ultraspectral stage when the spectral resolution reaches 1 nm. Hyperspectral and ultraspectral remote sensing are the frontier fields of remote sensing technology. It uses many very narrow electromagnetic wave bands to obtain relevant data from objects of interest so that substances that are not detectable in wide-band remote sensing can be detected. Chuanpeng Zhao et al. [43] proposed a new approach with solid improvements in dual-temporal image construction, misclassification processing, and tacit knowledge analysis, generated an accurate coastal salt marsh map at a national scale, and provided a classification mechanism for dual-temporal image-based coastal salt marsh identification and mapping. Remote sensing technology has huge advantages in obtaining real-time, large-scale, wide-area, and multi-period comparisons of basic coastal zone data. However, due to the indirect reflection of the characteristics of each observation object through the radiation and reflection characteristics of electromagnetic waves, the resolution is lower than that of conventional ground or ship observation methods. In addition, the presence of the phenomenon of homologous objects with different spectra and foreign objects with the same spectrum makes it difficult to recognize and interpret images.

Table 2. Satellite sensors information for marine research.

| Sensor Type | Function | Representative Satellite Sensors | Satellite | Country/Region | Time | Spectral Band | Resolution | Revisit Cycle |
|---|---|---|---|---|---|---|---|---|
| Seawater color sensor | Monitoring of marine surface chlorophyll concentration, suspended mass concentration, marine primary productivity, diffuse attenuation coefficient, other marine optical parameters [44] | Coastal Zone Color Scanner | NIMBUS-7 | United States | 1978 | 5 visible near-infrared bands (0.443~0.750 μm), 1 thermal infrared band (11.5 μm) | 0.825 km | 1 day |
| | | Sea-viewing Wide Field of View Sensor | Seastar | United States | 1997 | 8 bands (0.402~0.885 μm) | 1.1 km (IFOV) | 1 day |
| | | Moderate-resolution Imaging Spectroradiometer | EOS-AM(TERRA)/EOS-PM(AQUA) | United States | 1999/2002 | 36 discrete spectral bands (0.4~14.4 μm) | 0.25~1 km | 0.5/1 day |
| | | Chinese Ocean Color and Temperature Scanner | HY-1A HY-1C | China China | 2002 2018 | 10-band (0.402~12.50 μm) | 1.1 km 1.1 km | 1 day 1 day |
| | | Medium Resolution Spectral Imager | FY-3 | China | 2008 | 20-band (0.43~12.5 μm) | 0.25~1.1 km | 5.5 day |
| | | Chinese Ocean Color and Temperature Scanner | HY-1D | China | 2020 | 10-band (0.402~12.50 μm) | 1.1 km | 1 day |
| | | Ocean Colour Monitor | OceanSat-3 | India | 2022 | 3 bands | 360 m | 2 days |
| Infrared sensor | Measurement of marine surface temperature | Advanced Very High Resolution Radiometer | NOAA/TIROS | United States | 1979 | 2 thermal infrared channels (11 μm, 12 μm) | 1.1 km (IFOV) | 0.5/1 day |
| | | Along-Track Scanning Radiometer | ERS-1/2 | Europe | 1991/1995 | 2 thermal infrared channels (11 μm, 12 μm) | 1 km | 3 days (TIR)/6 days (SWIR) |
| | | Advanced Along-Track Scanning Radiometer | ENVISAT | Europe | 2002 | 11/12 μm channel during daytime and 3.7/11/12 μm channel at night | 1 km (IFOV) | 3 days (TIR)/6 days (SWIR) |
| | | VistA Integration Reporting and Revenue | FY-3A | China | 2008 | 10 channels (including visible channels, 3 infrared atmospheric window channels) | 1.1 km | 5.5 days |

**Table 2.** *Cont.*

| Sensor Type | Function | Representative Satellite Sensors | Satellite | Country/Region | Time | Spectral Band | Resolution | Revisit Cycle |
|---|---|---|---|---|---|---|---|---|
| Microwave altimeter | Measurement of mean sea level height, geoid, effective wave height, marine surface wind speed, surface laminar flow, gravity anomalies, rainfall index | Radar altimeter | ERS-1/2 | Europe | 1991/1995 | C-band/5.3 GHz | 20 km (IFOV) | 10 days/1 month |
| | | Dual frequency radar altimeter, Single frequency altimeter | Jason-1 | United States, France | 2001 | NRA: Ku-band/13.6 GHz and C-band/5.3 GHz; SSALT: Ku-band/13.65 GHz | 25 km | 10 days |
| | | Radar altimeter-2 | ENVISAT | Europe | 2002 | Ku-band/13.5 GHz | 20 km (IFOV) | 10 days/1 month |
| | | Radar altimeter | HY-2B | China | 2018 | Ku-band/13.58 GHz, C-band/5.25 GHz | 2 km | 14 days in the early stage, 168 days in the late stage |
| Microwave scatterometer | Measurement of the wind field at 10 m above sea level | Single-frame side-scan vertical transmit vertical receive (VV) radar | ERS1/2 | Europe | 1991/1995 | C-band/5.3 GHz | Optimal: 50 km; Sampling: 25 km | 3 days on average |
| | | Double amplitude side scan scatterometer | ADEOS | Japan | 1996 | Ku-band/13.995 GHz | Optimal: 50 km; Sampling: 25 km | 1.5 days |
| | | Sea Winds scatterometer | QuikSCAT | United States | 1999 | Ku-band/13.4 GHz | Optimal: 50 km; Standard: 25 km; Sampling: 12.5 km | 1 day |
| | | Microwaves scatterometer | HY-2B | China | 2018 | Ku-band/13.256 GHz | 25 km | 14 days in the early stage, 168 days in the late stage |
| | | Sector beam rotational scanning scatterometer | CFOSAT | China, France | 2018 | Ku-band/13.256 GHz | 25/12.5 km | real time |

| Sensor Type | Function | Representative Satellite Sensors | Satellite | Country/Region | Time | Spectral Band | Resolution | Revisit Cycle |
|---|---|---|---|---|---|---|---|---|
| Synthetic aperture radar | Monitoring of wave direction spectrum, mesoscale eddy, ocean internal waves, shallow sea topography, marine surface pollution, marine surface characteristic information [45] | Active Microwave Instrument Synthetic Aperture Radar | ERS-1/2 | Europe | 1991/ 1995 | L-band/1.275 GHz, S-band/3.0 GHz, C-band/5.3 GHz, X-band/9.6 GHz | 30 m | 1.5 month |
| | | Ka-band Radar | SWOT | United States | 2023 | Ku band/C band | / | 20.8 days |
| | | Multi-polarized Advanced Synthetic Aperture Radar | ENVISAT-1 | Europe | 2002 | C-band/5.3 GHz | 30 m~1 km | 5 days/3 months |
| | | Multi-polarized C-band synthetic aperture radar | GF-3 RCM-3 | China Canada | 2016 2019 | C-band/ C-band/5.405 GHz | 1 m (highest) 1 m (highest) | 1 week 12 days |
| Microwave radiometer | Measurement of marine surface temperature, marine surface wind speed, sea ice water vapor content, $CO_2$ and sea-air exchange | Line Leveling Passive Microwave Radiometer | DMSP | United States | 1987 | 4 bands (19.35, 2.235, 37.0, 85.5 GHz) 7 channels | 61 × 66 cm antenna diameter | 1 day |
| | | Microwave Scanning Radiometer | MOS-1 | Japan | 1987 | 2 bands (23.8 GHz, 31.4 GHz) | 23 km (31.4 GHz) /32 km (23.8 GHz) | 5 days |
| | | Multi-frequency scanning microwave radiometer | Seasat-A/ Nimbus-7 | United States | 1978 | 5 bands 9 channels | /22~120 km (37~6.6 GHz) | 2 days |
| | | Advanced Microwave Scanning Radiometer -E | EOS-PM(AQUA) | Europe | 2002 | 6 bands 12 channels | Antenna diameter 1.6 m | 1 day |
| | | Advanced Microwave Scanning Radiometer | ADEOS-2 | Japan | 2002 | 8 bands 14 channels | Antenna diameter 2 m | 1 day |
| | | Scanning Microwave Radiometer Imager MWRI | HY-2B | China | 2018 | 5 bands (6.925, 10.7, 18.7, 23.8, 37.0 GHz) | Antenna diameter 1.2 m | 14 days in the early stage, 168 days in the late stage |
| | | Calibration Microwave Radiometer | HY-2D | China | 2021 | 6 bands (3.58, 5.25, 13.256, 18.7, 23.8, 37 GHz) | / | 10 days |
| | | Poseidon-4 radar altimeter and microwave radiometer | Sentinel-6 | United States, Europe | 2020 | C-band/Ku-band | / | 10 days |

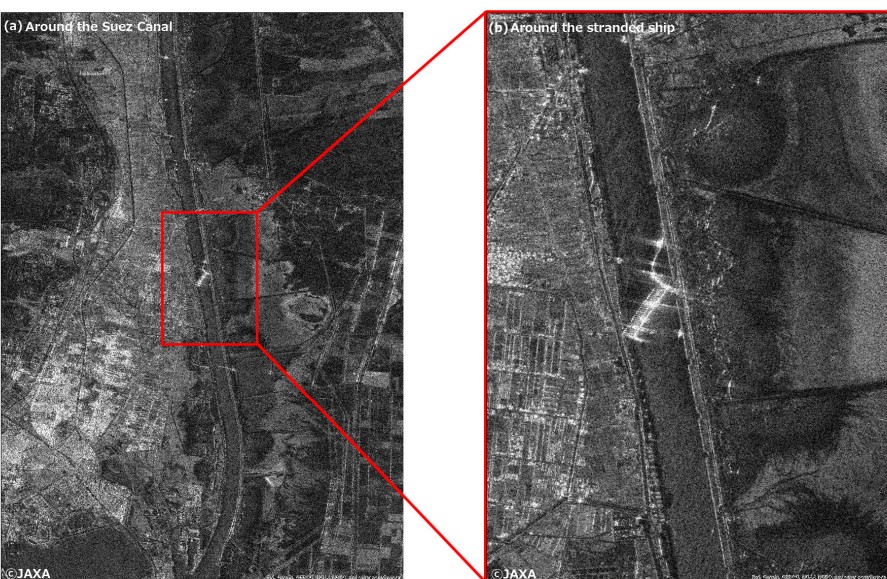

**Figure 7.** Image observed by PALSAR-2, synthetic aperture radar onboard ALOS-2, at 10:23 (UTC) on March 26 (**a**) around the Suez Canal; (**b**) extended image around the stranded ship (Picture source: Earth Observation Research Center).

### 3.2. The Monitoring Data and Methods

Marine remote sensing has the unique advantages of all-weather, wide-range, and long-term observation and is widely used in marine ecology and resources monitoring, marine disaster monitoring, marine rights and interests maintenance, marine environmental prediction and security assurance, and other fields. Therefore, marine pollution can be monitored and reversed based on marine remote sensing data. Marine pollution monitoring can be classified into three aspects: seawater quality, marine debris and microplastic pollution, and atmospheric deposition pollution. According to the application focus of remote sensing, the remote sensing data products and research methods for monitoring the pollution of these marine areas are analyzed.

#### 3.2.1. Seawater Quality Monitoring

Marine remote sensing is widely used in a variety of fields, including marine ecology and resource monitoring and marine disaster monitoring [46]. With the help of remote sensing, it is possible to reverse and monitor marine pollution. The main environmental indices monitored by remote sensing include the suspended solids content, chlorophyll-a concentration [47], colored soluble organic matter, and other comprehensive pollution indices [46,48]. To invert the quality of water environmental indices, a spectral model must be established and combined with monitoring data. Remote sensing inversion is often performed using visible light bands (Table 3), and various methods such as empirical models, theoretical models, semi-analytic models, and others can be used (Table 4). The suspended solids content in water can be reversed by fitting a formula that correlates the measured suspended solids content with remote sensing reflectance or turbidity [49–51]. The concentration of chlorophyll-a is a direct indices of eutrophication and organic pollution, and the inversion methods of chlorophyll-a concentration include chlorophyll fluorescence, band models, nonlinear mapping, and mechanism model methods [52–54]. Chromophoric soluble organic matter is another important component of dissolved organic matter that can be monitored remotely through the development of a remote sensing inversion model [55–57]. For example, red tides have posed a huge threat to the resources and environment of coastal areas. Multi-spectral scanners can be used to monitor red tides, whose visible/infrared multispectral radiometers can provide information on the location, range, water color type, changes in phosphate concentration on the sea surface, and the direction of red tide diffusion and drift to take timely measures to control them [58].

It is difficult to find independent spectral characteristics in seawater quality indices, such as dissolved organic carbon, water temperature [59], transparency, dissolved oxygen, chemical oxygen demand, five-day biochemical oxygen demand, total nitrogen, total phosphorus, etc. As a result, indirect remote sensing analysis must rely on the correlation between different substances [60–62]. However, remote sensing monitoring also has limitations, such as difficulty in estimating pollutants in the vertical dimension of water bodies and the limitation of partial inversion to estimate only general parameters, not specific types of marine pollution. Coastal human activities and the complex dynamic environment of the sea can also affect the water quality, requiring higher performance from satellite sensors to monitor it effectively through remote sensing technology [63]. Despite the progress in remote sensing technology, the main drawback of physical-chemical sensor systems is their lack of specificity and sensitivity and their inability to assess the environmental concentration of most marine pollutants. In this regard, biosensors may offer the required specificity and selectivity [64].

**Table 3.** The monitoring data for seawater quality.

| Seawater Quality Indices | | Data |
| --- | --- | --- |
| Suspended solids content | Remote sensing data | Sentinel-3A OLCI [65], SPOT, Terra ASTER, Landsat TM/ETM, Envisat MERIS [66], COMS GOCI [67–69] |
| | Measured data | Seawater sampling data, official water environment monitoring samples, measured spectral data |
| Chlorophyll-a concentration | Remote sensing data | Non-space remote sensing data: AVIRIS, CASI, OMIS, AISA, etc.; Remote sensing data: Envisat MERIS, CBERS-2 CCD, Terra/Aqua MODIS [70,71], EO-1 Hyperion, Terra ASTER, Landsat MSS/ETM+, HJ-1 CCD, et al. |
| | Measured data | Ground-measured spectral data: ASD, GER, etc. |
| chromophoric dissolvable organic matter | Remote sensing data | Sentinel-2 [72], Landsat 8 OLI, SeaWiFS, Aqua/Terra MODIS [73], etc. |
| | Measured data | Water sampling data |

**Table 4.** The monitoring methods for seawater quality.

| Methods | Characteristics |
| --- | --- |
| Empirical models (single band model, band ratio model, multiple regression model, nonlinear model, machine learning) | Based on the relationship between water spectral information and measured water quality parameters, the algorithm is relatively mature and the process is simple, but it lacks physical basis |
| Theoretical model | The combination of water quality spectral characteristics and statistical analysis has a certain physical basis, but for the model requiring higher precision, the process is more complicated and the universality is poor |
| Semi-analytical model | Based on the relationship between apparent optical quantity and measured water quality parameters, it has a strong physical mechanism and good universality, despite the difficulty of model establishment [72] |

The quality of seawater can be negatively impacted by large-scale pollutants such as red tides, green tides, and oil spills, especially due to increased human activities leading to increased inputs of nitrogen and phosphorus in the ecosystem (Table 5) [74–77]. Remote sensing technology plays a crucial role in monitoring such events by monitoring the seawater quality indices over large areas [78,79]. The research has shown that the multi-band

ratio method can be used to monitor floating algae and reduce interpretation errors [80–82]. Several algorithms have been proposed, including the Floating Algae Index (FAI) [83,84], the Normalized Algae Index (NDAI) [85], the Virtual Baseline Index of Floating Algae Height (VB-FAH) [86], and the Multi-Spectral Green Tide Index (MGTI) [87,88]. However, these algorithms require complex and accurate atmospheric correction procedures, which can increase the complexity of interpretation. To address this, Zhang Hailong et al. developed the Floating Algae Index (FGTI) based on the DN values of different satellite data, and Chen Ying et al. [89] proposed the Green Tide Index (TCT-GTI) algorithm based on the Tassel Cap Transformation method, which requires no atmospheric correction.

**Table 5.** The Monitoring data and methods for marine green tide pollution disaster.

| Author | Time | Satellite/Sensor | Waveband Range | Algorithm | Characteristics |
|---|---|---|---|---|---|
| Hu [78] | 2009 | Aqua/Terra MODIS | V, NIR, SWIR | Floating Algal Index | It is less sensitive to changes in environment and observational conditions (aerosol type and thickness, sun/observational geometry, and solar brilliance) and can penetrate thin clouds, providing a simple and effective means of atmospheric correction and making it easier to establish image-independent thresholds to monitor and quantify planktonic macroalgae. |
| Shi et al. [85] | 2009 | Aqua/Terra MODIS | SWIR | Normalized Algae Index | The effect of atmospheric molecular scattering is removed, making the NDAI more sensitive to the radiance signal from the marine surface [85]. |
| Son [90] | 2012 | GOCI | V, NIR | GOCI floating green tide index | The muted or subtle signal of planktonic green algae is enhanced and separated from the surrounding complex water signal [90]. |
| Xing et al. [86] | 2016 | HJ-1A/1B | V, NIR | Virtual Baseline Index of Floating Algal Height | Even without the use of the shortwave infrared (SWIR) band, VB-FAH appears to be comparable to the planktonic algal index (FAI). |
| Zhang et al. [88] | 2016 | GF1 WFV, HJ CCD | V | Multispectral Green Tide Index | It can effectively eliminate the influence of external interference, such as suspended sediment and thin clouds, and has low sensitivity to the environment [88]. |
| Zhang et al. [91] | 2019 | GF-1 WFV1, GF-1 WFV3, HJ-1B CCD, Landsat-7 ETM+, GOCI | V, NIR, SWIR | Floating algae index | The use of tasseled cap transformation is more powerful than traditional NDVI (normalized vegetation index) in responding to perturbations from environmental conditions, observational geometry, sunlight, and thin cloud pollution [88,91]. |
| Chen et al. [89] | 2020 | GOCI | V | Green Tide Index | No atmospheric correction is required [89]. |

Monitoring red tides can be conducted by using the spectral and temperature characteristics of the affected water bodies and through the analysis of optical satellite data (Table 6). The main methods of red tide remote sensing monitoring include the Chlorophyll Concentration Anomaly Method, the Red Tide Index (MRI), the Rrs Band Ratio Method, the Red Band Difference (RBD), the RBD_ KBBI (Karenia brevis Bloom Index), and others [92,93]. Jiang Dejuan et al. [92] emphasized that the effectiveness of red tide monitoring is dependent on the type of sensor, algorithm, and remote sensing time. Therefore, it is

necessary to combine field survey data or ERGB images with the remote sensing data to verify the accuracy of red tide monitoring in different times and regions [94,95].

**Table 6.** The monitoring methods for the marine red tide pollution disaster.

| Research Methods | Characteristics |
| --- | --- |
| Chlorophyll concentration anomaly method | It characterizes the most important characteristic parameters of red tide but is usually overestimated. |
| Red tide index | It can enhance the difference between the red tide water body and the surrounding water body; therefore, it can be used for the determination of red tide. |
| Band ratio method | Using the ratio of reflection and absorption bands of red tide water bodies, we can extract red tide information. |
| Red band difference method | The method is based on the high fluorescence property of dinolflagellate water bloom. |
| RBD_ KBBI | The red tide monitoring index proposed is based on RBD. |

Note: Compiled from references [92].

Marine engineering encompasses a variety of activities, including marine dumping, marine oil and gas transportation, and others. The main source of seawater quality and environmental degradation in the ocean is oil spills caused by ships during berthing at ports, navigation, and accidents [96,97]. Early monitoring, tracking, and diffusion of oil spills is crucial in designing effective algorithms for remote sensing monitoring [98]. However, the current methods for oil spill monitoring often overlook the space-time characteristics and laws of the interaction between oil spills and the ocean [99]. Different remote sensing technologies, such as microwave radar, optical (multi/hyperspectral remote sensing), thermal infrared, and others, can be used to monitor marine oil spills. Microwave radar remote sensing is one of the main methods and is based on the different absorption and transmission properties of water and oil in electromagnetic waves. This method is mainly used to extract the oil spill extent by identifying the "dark pixel" feature in the microwave radar image [100,101]. Spectral response characteristics of oil spill simulation experiments can be used to identify, classify, and quantify the types of oil spills [102,103]. Optical remote sensing can also be used for the identification and classification of oil spills [104–106]. Solar reflected light can be used to monitor the oil spill thickness on the marine surface; however, the monitoring range is limited to less than 0.4 mm due to the different absorption, scattering, and reflection effects of oil spills on incident light. Thermal infrared remote sensing has a larger monitoring range; however, the ability to monitor oil spills is weaker than that of optical remote sensing due to the differences in existing spaceborne sensors [107]. In a study of oil spills caused by marine accidents, Lu Yingcheng et al. [105] used a GF-3 SAR image to delineate "suspected oil spills" and analyzed the optical anomalies of the "Sanji" oil spill incident in the East China Sea in 2018 using Sentinel-2 multi-spectral remote sensing data (MSI) from the European Space Agency. This study resulted in the optical remote sensing identification and classification of different types of oil spills. Huang Ke et al. [104] used GF-1 satellite data along with a thin oil film thickness model to calculate the oil film thickness of the oil spill area and estimate the oil spill caused by the explosion of the Huangwei oil pipeline in Qingdao in 2013. Remote sensing monitoring of oil spills at sea is influenced by marine environmental conditions, such as marine surface wind speed, and therefore, a combination of sea wave spectrum and wind wave information is necessary for further research [108,109].

### 3.2.2. Debris and Microplastics Monitoring

Marine debris and microplastics have a significant negative impact on marine organisms, ecosystems, fisheries, and tourism [110,111]. The monitoring of these pollutants in the ocean through remote sensing can be effectively accomplished by using sensitivity analysis, optical simulation, and satellite image spectral analysis (Table 7) [112,113]. The reflection spectra of known marine floating objects can be utilized for this purpose [110,114]. Most

of the research and monitoring of marine debris is focused on plastic waste and utilizes optical sensors operating in the visible light, near-infrared, and short-wave infrared spectral ranges, as well as SAR sensors and other image data sources [115]. However, microplastics are plastic pollutants with a diameter of less than 5 mm and cannot be effectively identified through satellite images due to their small size. While there have been efforts to observe microplastics In water, through optical means such as lasers and polarization, there is still a significant gap in terms of satellite observation. SAR sensors are commonly used to monitor marine debris and microplastics by identifying surface active agents on the marine surface. The classification methods for marine debris monitoring are mainly manual classification, use of indices, spectral classification, or machine learning [115]. The use of unmanned aerial vehicles and artificial intelligence for monitoring beach litter is an emerging field [116]. Machine learning has been shown to significantly increase the speed of screening for marine debris, operating 39 times faster than human screening in cases of low sensitivity [116]. The development of a spectral characteristic model for marine debris or microplastics is also an area of exploration. Goddijn-Murphy et al. proposed a reflection model based on geometric optics and the spectral characteristics of plastics and seawater to describe the interaction between sunlight and floating microplastics on the marine surface.

The use of satellite data are useful tool in monitoring marine debris as it can provide repeated global coverage data at various scales and resolutions, which is not possible through field observations or remote sensing with ships, aircraft, or unmanned aerial vehicles. However, the limitations of optical satellite data include limited temporal coverage due to fixed time and factors such as the sun, clouds, atmospheric aerosols, sensor saturation on surfaces such as ice, sand, or snow, and a high solar zenith angle. Additionally, the spatial resolution of satellite data are typically greater than 1 m, making it difficult to monitor marine debris smaller than this resolution [117]. For the limitation, it is possible to solve this problem by improving the accuracy of remote sensing observation instruments. Furthermore, through multi-band remote sensing observation methods, we can gain a deeper understanding of the electromagnetic wave characteristics of ground objects and separate information from different observation objects.

**Table 7.** Remote sensing monitoring data and methods for marine debris and microplastics.

| Author | Time | Band Range | Satellite/Sensor | Types of Plastic Waste | Monitoring Methods | Classification Methods |
|---|---|---|---|---|---|---|
| Hamilton [117] | 2015 | MW | RADARSAT-2 | Marine surface float (marine bacteria associated with surfactants) | Observe oil slicks to monitor marine surface slicks using the association among bacteria produced by surfactants and oil slicks (Hamilton et al., 2015) | Machine learning techniques |
| Aoyama [118] | 2016 | NIR/SWIR | Worldview-2 | Marine debris | Find common spectral features associated with the presence of plastic [118] | Spectral classification method |
|  | 2016 | NIR/SWIR | Worldview-3 | Buoys around fishing nets | Propose the spectral angle mapper (SAM) classification method [118] | Spectral classification method |
| Kurata [119] | 2016 | MW | RADARSAT-2 | Marine surface float (marine bacteria associated with surfactants) | Observe oil slicks to monitor marine surface slicks using the association among bacteria produced by surfactants and oil slicks [119] | Machine learning techniques |
| Davaasuren [120] | 2018 | MW | Sentinel-1A, COSMO-SkyMed | Microplastics (surfactants, sea mud and biofilms) | Observe oil slicks to monitor marine surface slicks using the association between bacteria produced by surfactants and oil slicks [120] | Machine learning techniques |
| Goddijn-Murphy | 2018 | V~SWIR | -- | Floating microplastics | Model the reflection of sunlight interacting with the marine surface of floating microplastics | Model building |
| Howe [121] | 2018 | MW | TerraSAR-X | Marine surface float (marine bacteria associated with surfactants) | Observe oil slicks to monitor marine surface slicks using the association between bacteria produced by surfactants and oil slicks [121] | Machine learning techniques |
| Topouzelis [115] | 2019 2019 | V, NIR MW | Sentinel-2A Sentinel-1 | Artificial floating plastics (water bottles, LDPE plastic bags and nylon fishing nets) | Supervised classification [108] | Machine learning techniques |
| Biermann | 2020 | V, NIR, SWIR | Sentinel-2 MSI | Piece of floating large plastic | Develop novel float index (FDI) algorithms to monitor floating plastics and distinguish them from natural floats | Usage index |
| Kikaki [111] | 2020 | V, NIR NIR, SWIR NIR, SWIR | Planet P Sentinel-2 MSI Landsat-8 OLI | Floating plastic debris | Monitor and validate floating plastic debris by systematically recording and assessing the spectral characteristics of pure floating plastic and distinguishing it from other floating materials on the marine surface (e.g., Sargassum, foam) | Manual classification |

### 3.2.3. Inversion of Atmospheric Pollution Deposition

The monitoring of atmospheric pollution deposition typically involves measured data, model simulations, and remote sensing. Atmospheric pollution deposition in the ocean can be classified into three sub-categories that are influenced by the distribution of aerosols: clean ocean, marine minerals, and marine pollution [122]. Therefore, the monitoring of atmospheric pollution deposition can be conducted through aerosols. Remote sensing by satellite is a highly effective research method for observing the global distribution of aerosols, their optical properties, and their radiation effects due to the large spatial and temporal variations of aerosol distribution [123]. Aerosol optical thickness, a key parameter of aerosols, characterizes atmospheric turbidity and is a critical factor in determining the impact of aerosols on climate [124,125]. Some of the satellite sensors used to monitor atmospheric pollution deposition include the Global Ozone Monitoring Experiment (GOME), the Atmospheric Mapping Scanning Imaging Absorption Spectrometer (SCIAMACHY), the Ozone Monitor (Aura OMI), the Interferometric Infrared Atmospheric Detector (METOP/IASI), the FY-3 Total Ozone Detector (FY-3 TOU), and the UV Ozone Vertical Detector (FY-3 SBUS) of FY-3 [126,127].

Remote sensing monitoring calculates the concentration of deposition according to the formula by reversing the density of the vertical gas column, and then calculates the dry and wet deposition flux in combination with the dry and wet deposition calculation formula. For example, Dong Haiying et al. [124] used Terra MODIS data to reverse the 10 km resolution aerosol optical thickness and analyzed that the aerosol concentration gradually decreased from the coastal waters to the outer sea due to the influence of land-based pollution components, and there was a high aerosol optical thickness in the coastal Bohai Sea, the Yellow Sea coast, and the Yangtze River estuary. SeaWiFS aerosol optical thickness products can be used to study aerosol distribution and change characteristics. Hao Zengzhou et al. [128] analyzed the aerosol distribution in China's sea area and found that the average aerosol optical thickness in the eastern China sea area has a zonal distribution centered on the middle latitude and has seasonal changes. From spring to winter, the aerosol optical thickness shifts from the high latitude sea area to the low latitude sea area, and the scope is also gradually expanding. Mao Ying [122] over the sea air will be visible infrared imaging radiometer VIIRS, medium resolution imaging spectrometer MODIS, stationary seawater color satellite imager GOCI, and geosynchronous meteorological satellite AHI H8 aerosol optical thickness product data combined with field measured data. Compared with other aerosol optical thickness remote sensing products, GOCI aerosol optical thickness products show the characteristics of high sampling frequency and high precision.

The inversion results of aerosol optical thickness remote sensing products are affected by a variety of factors, including the signal-to-noise ratio of the sensor itself, the accuracy of the calculation of the surface reflectance of the underlying surface, and the rationality of the selection of the preset aerosol model [122]. At the same time, the frequent human activities and pollutant emission and diffusion, as well as the unique physical geography and hydrological conditions of the sea area, will cause the sea area reflectivity to include many factors, not only the atmospheric impact but also the role of the seawater itself, which is more obvious in the coastal sea area.

## 4. Conclusions and Future Prospects

### 4.1. Conclusions

This paper summarizes the progress in the application of remote sensing for marine pollution monitoring. Coastal areas are subjected to various forms of pollution from land, sea, and air sources due to the difference in urbanization and industrialization, leading to a degradation in the quality of the marine environment and having significant impacts on marine ecology. Remote sensing monitoring of marine pollution encompasses various aspects, including seawater quality, marine debris and microplastic pollution, and the inversion of atmospheric pollution depositions. With the launch of various marine monitoring satellites, remote sensing provides a wealth of data sources to monitor marine hydrology

and environmental parameters. Remote sensing can provide information on environmental parameters and habitats and monitor human activities, thus improving the marine environment to some extent (Figure 8). Monitoring seawater quality and temperature mainly uses remote sensing of seawater color and temperature, which is effective in monitoring seawater quality deterioration and large-scale coastal seawater quality changes during disasters. The methods of monitoring marine debris and microplastics have evolved from remote sensing inversion to machine learning. The research into remote sensing monitoring of marine atmospheric pollution deposition, which only quantifies atmospheric pollution through aerosol thickness, is inadequate. It is crucial to integrate measured data, remote sensing monitoring, and information services to achieve comprehensive monitoring and assessment of marine pollution. Moreover, there are a number of marine remote sensing products worldwide; however, there are significant differences in accuracy and other aspects of similar products in different countries, leading to technical barriers that pose challenges to marine governance decisions and environmental protection in various countries and regions. Therefore, developing three-dimensional monitoring techniques, combining different types of sensors, and establishing a comprehensive marine monitoring database is of great significance, along with data, technology, and standard sharing, to monitor the marine environment and realize the sustainable development of marine resources.

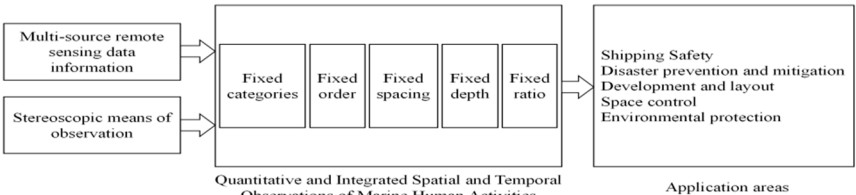

**Figure 8.** The extraction process of marine pollution data. Note: Redraw from reference [8].

*4.2. Future Prospects*

4.2.1. The Tendency of Three-Dimension in Marine Monitoring

Despite its practicality, remote sensing monitoring has limitations in its ability to identify and quantify the majority of marine pollutants. Due to the complex three-dimensional nature of the ocean, most remote sensing methods can only obtain information from the upper layer, with the exception of the altimeter used for rough sounding. The best conditions for optical sensors to penetrate the seawater surface are up to 27 m, while air sensors can only reach up to 46 m [129,130]. This resolution limitation of the image data affects the vertical dimension of marine pollution monitoring and assessment and cannot provide a quantitative method for tracing the levels of individual substances. Thus, the development of remote sensing technology that can monitor deep-sea pollution is a crucial goal for the future [117,131]. Although monitoring the deep sea may monitor less pollution than the upper sea, three-dimensional marine observation remains the key direction for the progress of remote sensing technology. Furthermore, multi-source data integration with three-dimensional ocean numerical models for marine pollution monitoring should be used. Observations obtained through remote sensing and sensors are not sufficient to project the 3D structure of the marine environment. Hence, 3D hydrodynamic and ocean circulation modeling should be developed.

4.2.2. The Tendency of Multi-Source Data Fusion in Marine Pollution Monitoring

Despite the progress in remote sensing technology and the increasing availability of data, the accuracy of remote sensing data products can still be impacted by environmental factors such as spatial-temporal variability in the sky and on earth. The diversity of human activities in the ocean also limits the validity of remote sensing data for marine pollution monitoring. Currently, marine pollution monitoring primarily relies on field observations, utilizing biochemical indices, rather than solely relying on remote sensing. To improve the accuracy and reliability of marine pollution monitoring, it is necessary to integrate remote

sensing with other monitoring tools, such as ship sensors, buoy sensors, aircraft sensors, and marine ecological reserves/biosensor networks.

Remote sensing technology, such as high-resolution imaging, multi-spectral and hyper-spectral, fluorescence, and Raman molecular spectroscopy, can be used to measure different types of pollution; however, it is still not enough to replace on-site observations. For example, the movement of pollutants in the ocean is influenced by complex hydrodynamics and marine circulation, making it difficult for remote sensing technology to accurately track the diffusion of marine pollution. Additionally, micropollution may not cause large-scale pollution, and it is challenging to quantify using remote sensing alone.

Therefore, multi-platform and multi-means integrated observation technology is necessary for accurate marine environmental monitoring. This includes the establishment of a marine remote sensing database, which integrates satellite, aviation, ship, and shore-based data to provide a comprehensive monitoring system for marine environmental monitoring. The development of technology for collecting, storing, processing, analyzing, and utilizing multi-source data will be crucial for the progress of marine remote sensing. By providing timely information to the government and relevant entities, this database can support disaster prevention, mitigation, and the launch of emergency plans.

In addition, the importance of ground measurements and alternative satellite calibration systems (e.g., MOBY, AERONET, and GOOS) cannot be overlooked. The Marine Optical Buoy (MOBY), a radiometric buoy stationed in the waters off Lanai, Hawaii, has been the primary basis for the on-orbit vicarious calibrations of all three US ocean color sensors and numerous international satellite sensors since late 1996 [132]. The aErosol rObotic NETwork (AERONET) program is a federation of ground-based remote sensing aerosol networks established by NASA and PHOTONS. For more than 25 years, the project has provided a long-term, continuous, and readily accessible public domain database of aerosol optical, microphysical, and radiative properties for aerosol research and characterization, validation of satellite retrievals, and synergism with other databases. The Global Ocean Observing System (GOOS) consists of a variety of observation means and is committed to obtaining and disseminating reliable assessment and forecasting information on the present and future state of the marine environment for the effective, safe, and sustainable use of the marine environment.

### 4.2.3. The Benefit for the Sustainable Utilization of Marine Resources

The relationship between marine resources and the environment determines the sustainable utilization of the ocean, and remote sensing technology for monitoring marine resources and the environment can enable the effective implementation of marine spatial planning. In the governance of marine resources, remote sensing methods are used to strengthen the survey of the reserves and distribution of various marine resources and monitor the impact of human activities on marine development and environmental pollution. Based on the principle of unified coordination of economic, social, and ecological benefits, by studying the intensity and methods of human utilization of the ocean, we improve marine spatial planning, ensure the rational use of marine resources, and promote sustainable development of the marine economy. In addition, comprehensive monitoring data based on remote sensing databases is also the decision-making basis for marine scientific utilization.

**Author Contributions:** Conceptualization, J.M. and R.M.; methodology, Q.P. and X.L.; software, Q.P. and J.W.; valida-tion, Q.P. and X.N.; formal analysis, J.M. and X.L.; investigation, Q.P. and J.W.; resources, R.M.; visualization, Q.P. and X.L.; supervision, R.M.; funding acquisition, J.M. All authors have read and agreed to the published version of the manuscript.

**Funding:** This research was supported by Key Laboratory of Ocean Space Resource Management Technology, MNR (Grant No. KF-2023-103), the Natural Science Foundation of Zhejiang Province, China [Grant No. LGF22D010002], the Open Funding of Zhejiang Collaborative Innovation Center

for Land and Marine Spatial Utilization and Governance Research [Grant No. LHGTXT-2023-003], and Ningbo University Donghai Academy [Grant No. DH202302CBZZ03].

**Data Availability Statement:** Not applicable.

**Conflicts of Interest:** The authors declare no conflict of interest.

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
