# Peer review of "A Global Review of Progress in Remote Sensing and Monitoring of Marine Pollution"

_water, doi:10.3390/w15193491_

Round 1
Reviewer 1 Report
A fruitful review that summarises selected publications and trends in the use of remote sensing marine pollution. The author collective has diversified the key areas of application of remote sensing in marine pollution monitoring. I appreciate the elaboration of the accompanying tabular appendices, which summarise institutions, technologies or trends selected for discussion in the text. The conclusion is also well done.
Although the authors' team has focused on three aspects of marine pollution, the introductory passage and chapter 2 should provide a complete and comprehensive overview of the sources and monitored indicators of marine pollution (physical, chemical, biological and thermal). I would, therefore, welcome a mention of underwater noise pollution or thermal pollution of coastlines. Due to the nature of the article (review), it is not necessary to expand the other chapters, it is only required to mention all relevant aspects of marine pollution, even regarding the title. The recommendation given is not strictly necessary. The proposal is for consideration.
For review-type papers, it is necessary to accept the broadest possible platform of sources; I am missing citations from the International Journal of Remote Sensing.
Isolated typos and omissions can be found in the article:
Page 8 - hyperspectral and hyperspectral should be ultraspectral
Page 17 - deposion - text size
Table 2 has poorly managed formatting, and categorising into groups in columns and rows is not legible. The comment relates to the Function and R.S. Sensors columns.
All cited sources are listed twice in the references. The number of citations is then significantly reduced.
The project affiliation is unfinished at the end of the text.
Author Response
Dear Prof. and Editor
Many thanks to the reviewers for their comments and suggestions. These have helped us to refine the paper, in line with their comments, as we explain below.
Highlighted in yellow here are the comments we regarded as central to their reports. We have attempted to respond to all these points; our answers are in italics. In the revised text itself, all revisions are tracked in this paper.
Our main revisions are divided into three parts: Firstly, isolated typos, omissions and the format issues have been revised in the article. Secondly, the paper were deepened to provide better clarity and information for international readers. Third, the English expression of the paper was optimized.
We trust the revisions to the manuscript will satisfy our colleagues.
With best wishes
Reviewer #1:
Point 1: hyperspectral and hyperspectral should be ultraspectral.
Response 1: Thanks for your suggestion. We have revised this typo.
Point 2: deposion - text size.
Response 2: Thanks for this suggestion. We have revised this typo.
Point 3: Table 2 has poorly managed formatting, and categorising into groups in columns and rows is not legible. The comment relates to the Function and R.S. Sensors columns.
Response 3: Thanks for your suggestion. We have adjusted the Table 2.
Point 4: All cited sources are listed twice in the references. The number of citations is then significantly reduced.
Response 5: Thanks for your suggestion. We have delete the repetitive references.
Great thanks for your professional suggestions, we have go through the paper and revised the isolated typos and omissions you have raised in the paper, and some other typos and omissions also have been revised. In the next step, we will also have further research on the marine environment issues based on the remote sensing, such as underwater noise pollution and thermal pollution of coastlines, and the paper is a part of our monographic research.

Reviewer 2 Report
Attached in the file

Author Response
Dear Prof. and Editor
Many thanks to the reviewers for their comments and suggestions. These have helped us to refine the paper, in line with their comments, as we explain below.
Highlighted in yellow here are the comments we regarded as central to their reports. We have attempted to respond to all these points; our answers are in italics. In the revised text itself, all revisions are tracked in this paper.
Our main revisions are divided into three parts: Firstly, isolated typos, omissions and the format issues have been revised in the article. Secondly, the paper were deepened to provide better clarity and information for international readers. Third, the English expression of the paper was optimized.
We trust the revisions to the manuscript will satisfy our colleagues.
With best wishes
Reviewer #2:
Point 1: The introduction should provide a more comprehensive background on the extent and impact of marine pollution caused by urbanization and industrialization. This will help readers understand the urgency and significance of the issue.
Response 1: Thanks for your suggestion. We have added related introduction in the paper, and we hope it will help the international readers understand the urgency and significance of the issue. The following is the added content: generally, the coastal natural shoreline and coastal mudflat wetlands have been continuously reduced, the area of mangroves and coral reefs has been greatly reduced, and the marine ecological environment has been polluted. The situation is becoming increasingly serious, with increasing eutrophication of seawater, frequent occurrences of marine ecological disasters such as brown tides, green tides, and red tides, posing a serious threat to migratory waterfowl and marine biodiversity.
Point 2: The section on remote sensing technology should provide a more detailed explanation of how it works and why it is an effective tool for monitoring marine pollution. This will help readers understand the advantages and limitations of remote sensing in this context.
Response 2: Thanks for this suggestion. We have provide a more detailed explanation, and we hope it will help readers understand the advantages and limitations of remote sensing in this context. The following is the explanation we provided: Remote sensing technology has huge advantages in obtaining real-time, large-scale, wide-area, and multi-period comparisons of coastal zone basic data. However, due to the indirect reflection of the characteristics of each observation object through the radiation and reflection characteristics of electromagnetic waves, the resolution is lower than that of conventional ground or ship observation methods. In addition, the presence of the phenomenon of homologous objects with different spectra and foreign objects with the same spectrum makes it difficult to recognize and interpret images.
Point 3: The section on seawater pollution should include specific examples of how remote sensing technology has been used to monitor and assess this type of pollution. This will provide concrete evidence of its effectiveness and contribute to the overall credibility of the manuscript.
Response 3: Thanks for your suggestion. We have provided specific examples as follows: For example, red tides have posed a huge threat to the resources and environment of the coastal areas. Multi-spectral scanners can be used to monitor red tides, whose visible/infrared multispectral radiometers can provide information on the location, range, water color type, changes in phosphate concentration on the sea surface, and the direction of red tide diffusion and drift to take timely measures to control them.
Point 4: The section on marine debris and microplastic pollution needs further development. It should include a discussion of current monitoring methods and their limitations, as well as potential approaches that can be explored in future research. This will ensure a more comprehensive coverage of the topic.
Response 4: Thanks for your suggestion. The section has been further developed as follows: The use of satellite data is a useful tool in monitoring marine debris as it can provide repeated global coverage data at various scales and resolutions, which is not possible through field observations or remote sensing with ships, aircraft, or unmanned aerial vehicles. However, the limitations of optical satellite data include limited temporal coverage due to fixed time and factors such as the sun, clouds, atmospheric aerosols, sensor saturation on surfaces such as ice, sand, or snow, and high solar zenith angle. Additionally, the spatial resolution of satellite data is typically greater than 1 meter, making it difficult to monitor marine debris smaller than this resolution (Hamilton et al, 2015). For the limitation, the methods are possible to solve this problem by improving the accuracy of remote sensing observation instruments. Furthermore, through multi band remote sensing observation methods, we can gain a deeper understanding of the electromagnetic wave characteristics of ground objects and separate information from different observation objects.
Point 5: The section on marine air pollution should provide a clearer explanation of the specific air pollution parameters that are currently monitored and the challenges associated with accurate monitoring. This will help readers understand the gaps in current methods and the potential for improvement.
Response 5: Thanks for your suggestion. In this section, we has provided the explanation as follows: The inversion results of aerosol optical thickness remote sensing products are affected by a variety of factors, including the signal-to-noise ratio of the sensor itself, the accuracy of the calculation of the surface reflectance of the underlying surface, and the rationality of the selection of the preset aerosol model (Mao Ying et al., 2021). At the same time, the frequent human activities and pollutant emission and diffusion, as well as the unique physical geography and hydrological conditions of the sea area, will cause the sea area reflectivity to include many factors, not only the atmospheric impact, but also the role of the seawater itself, which is more obvious in the coastal sea area.
Point 6: The conclusion should summarize the main findings of the manuscript and provide clear recommendations for future research. Specifically, it should highlight the importance of developing three-dimensional monitoring techniques, combining different types of sensors, and establishing a comprehensive marine monitoring database.
Response 6: Thanks for your suggestion. We have revised this part in accordance to the suggestion.
Point 7: There are lots of commas and full stops are missing.
Response 7: Thanks for your suggestion. We have revised these issues. For example, in Table 2, there are not full stop previously, and we have added the full stop now.
Point 8: references : There are fewer numbers of new references or recent referencesespecially related to ground-based to compare satellite observation studies. Therefore.for current study should be given a strong impact if you can cite the following reference.
Response 8: Thanks for your suggestion. We have added these references.
Great thanks for your professional suggestions, we have go through the paper and revised the problems you have raised in the paper, and some other typos and omissions also have been revised. In the next step, we will also have further research on the marine environment issues based on the remote sensing.
